# Emerging Concern with Imminent Therapeutic Strategies for Treating Resistance in Biofilm

**DOI:** 10.3390/antibiotics11040476

**Published:** 2022-04-02

**Authors:** Ramendra Pati Pandey, Riya Mukherjee, Chung-Ming Chang

**Affiliations:** 1Centre for Drug Design Discovery and Development (C4D), SRM University, Delhi-NCR, Rajiv Gandhi Education City, Sonepat 131 029, Haryana, India; ramendra.pandey@gmail.com (R.P.P.); riya.mukherjee1896@gmail.com (R.M.); 2Master & Ph.D. Program in Biotechnology Industry, Chang Gung University, No.259, Wenhua 1st Road, Guishan District, Taoyuan 33302, Taiwan

**Keywords:** biofilm, biofilm infection, biofilm resistance, nanoparticles, abiotic and biotic factors

## Abstract

Biofilm production by bacteria is presumed to be a survival strategy in natural environments. The production of biofilms is known to be influenced by a number of factors. This paper has precisely elaborated on the different factors that directly influence the formation of biofilm. Biofilm has serious consequences for human health, and a variety of infections linked to biofilm have emerged, rapidly increasing the statistics of antimicrobial resistance, which is a global threat. Additionally, to combat resistance in biofilm, various approaches have been developed. Surface modifications, physical removal, and the use of nanoparticles are the recent advances that have enabled drug discovery for treating various biofilm-associated infections. Progress in nanoparticle production has led to the development of a variety of biofilm-fighting strategies. We focus on the present and future therapeutic options that target the critical structural and functional characteristics of microbial biofilms, as well as drug tolerance mechanisms, such as the extracellular matrix, in this review.

## 1. Introduction

A biofilm is a community of microorganisms, especially bacteria, that maintains an organized and structured strategy for growing and proliferating on any surface for their survival. The organization of the cells among the bacteria in the biofilm has primarily been revealed by examining single species. The survival of the bacteria present in the biofilm is feasible because of their orientation in the form of microcolonies, which are encapsulated in the extracellular polymeric substance of the matrix. These are separated by open water channels that serve as the primordial circulatory system for the transportation of nutrients and the disposal of metabolic waste products [1]. All the specific bacteria maintain a microenvironment controlling the pH, nutrient availability, and temperature, which influence the biofilm growth. Biofilm maturation is a multi-stage developmental process with distinct characteristics that should be taken into account when developing antibiotic treatment regimens for biofilms. Biofilms have been extensively accepted as one of the main causes responsible for human diseases [2].

Quite a large number of diseases arise only from biofilm infections [3]. Numerous antibiotics are prescribed against these infections, but increased antibiotic resistance has proven to be a common trait associated with biofilm bacteria [4]. This ineffectiveness of antibiotics against biofilm-associated infections has steadily increased the range of antimicrobial resistance (AMR). When novel resistance mechanisms evolve, AMR affects the efficient prevention and treatment of diseases caused by infection-causing microbes, such as bacteria, fungi, and viruses [5]. Long-term disease, disability, and deaths have been encountered from emerging AMR effects. According to estimates [6], AMR is expected to kill over ten million people by 2050. Antibiotics are becoming ineffective due to the fast rise of multidrug-resistant microorganisms. Among the many variables that contribute to AMR, intrinsic biofilm growth has been identified as a critical component [7]. Furthermore, antibiotic tolerance developed from biofilms leads to dangerous recurrent chronic infections. As a result, the discovery of a novel strategy is a potential solution that can aid in the fight against AMR. To combat such a problem, an intervention using the One Health Approach is highly encouraged among the researchers for a quicker and better solution [8].

Additionally, nanotechnology is a promising technology in the medical profession that can be used in a variety of therapeutic settings. The potential use of nanoparticles as antimicrobial agents is being investigated and considered as an alternative to address the difficulties faced by healthcare personnel and patients in preventing infections caused by pathogenic bacteria. Bacterial biofilms and other microorganisms are commonly found on medical devices and wounds, where their contents disseminate and cause illnesses. Antimicrobial-loaded nanoparticles, alone or in combination with other compounds, could improve the bactericidal activity of nanomaterials, inhibiting biofilm formation and overcoming antibiotic resistance [9].

This includes effectively destroying infections without hurting other cells or producing any negative impact on living cells. In this review, we have summarized the globally emerging concern about biofilm resistance and their prevention by discussing some of the prominent therapeutic strategies combating it. 

## 2. Pathogen

### 2.1. Bacterial Type and Importance of One Health Approach

In the formation of biofilm, the involvement of microorganisms is diverse and takes many forms. Various types of microbes play a vital role in the formation of biofilm. As a consequence of this, the type of pathogen is a prime factor that controls the biofilm formation effectively. Together, both Gram-positive and Gram-negative microorganisms participate in biofilm formation [10]. Bacteria usually show the ability to bind with substrates due to high adherence potential, and they form biofilms that manifest two kinds of living states: one is a free-living, planktonic living state and the other one is a sessile living state [11]. Some bacterial types have the advantage of forming a biofilm, as this phenomenon is considered to be one of the methods for microbial self-defence. This formation, in due course, helps the cells of the biofilm to remain in a favourable condition that creates a niche in which they exist. Bacterial cells are thought to grow naturally in close proximity to one another [12]. An innumerable number of bacteria, including Gram-positive and Gram-negative bacteria, control the formation of biofilm. Gram-positive bacteria include *Bacillus* spp., *Listeria monocytogenes*, *Staphylococcus* spp., and lactic acid bacteria, such as *Lactobacillus plantarum* and *Lactococcus lactis*. Gram-negative bacteria include *Escherichia coli*, *Klebsiella pneumoniae*, *Pseudomonas aeruginosa*, and *Acinetobacter baumannii*, which are responsible for various pathological diseases [13]. Biofilm formation is very prominent in implant-associated infections as well. The majority of the implant-associated infections emerge from biofilm formation, which is difficult to treat with the conventional methods of antibiotics. *Staphylococcus aureus* and *Streptococcus oralis* are Gram-positive bacteria found to cause implant-associated infections. *Pseudomonas aeruginosa* and *Aggregatibacter actinomycetemcomitans* are Gram-negative bacteria that play a vital role in generating implant-associated infections [14]. The below figure (Figure 1) gives some information about the different microorganisms, along with the biofilm-associated diseases that they are capable of forming [15]. Antimicrobial resistance genes circulate across the microbiomes of humans, animals, and the environment, which make up the One Health concept’s many sectors. One Health is defined as a multidisciplinary effort—working locally, nationally, and globally—to achieve optimal health for people, animals, and the environment through policy, research, education, and practice (3). The One Health concept has previously focused on the interconnections and interdependencies among sectors at local sites; however, in light of global health, the understanding of communication among local ecosystems and the identification of factors that contribute to the global AMR crisis have recently received a lot of attention (3). The human microbiomes of the gut, skin, and respiratory tract have their resistomes (a collection of all antimicrobial resistance genes) which are assessed using metagenomics based on next-generation sequencing technologies. Controlling antimicrobial resistance gene flow from other sectors to the human sector, particularly antimicrobial resistance gene transfer to disease-causing bacteria, requires an understanding of the dynamics of the human resistome (the collection of all antimicrobial resistance genes) and its relationships with the other One Health sectors [16,17]. Removing contaminated foreign bodies, the selection of biofilm-active, sensitive, and well-penetrating antibiotics, topical antibiotic delivery in high doses and combinations, and anti-quorum sensing or biofilm dispersion agents are all elements of a multidisciplinary partnerships that are taken care of by the One Health Approach [16].

### 2.2. Genetic Regulation of Biofilm Formation

In Gram-negative organisms, the genetic regulators of biofilm formation include quorum sensing (QS), bis-(3′-5′)-cyclic diguanosine monophosphate (c-di-GMP), and small RNAs (sRNAs). Quorum sensing (QS) is regarded as an important regulator for intercellular communication. QS is entirely based on small, self-contained, signal-generating molecules known as autoinducers. When there are enough bacteria existing and the accumulation of autoinducers attains a critical mass, the bacteria sense this and respond by suppressing or activating target genes. QS systems are important in the formation and dispersal of bacterial biofilms. Even though these systems are not engaged in the attachment and early stages of biofilm growth, they are required for subsequent biofilm development and are the primary regulators of biofilm dispersal. During the formation and spread of bacterial biofilms, QS systems are extremely crucial. The c-di-GMP signalling network, the second key biofilm regulator, is the most complex secondary signalling system identified in bacteria. Selection among planktonic and biofilm-associated lifestyles is influenced by c-di-GMP [18]. The synthesis of exopolysaccharides, sticky pili, and adhesins, the secretion of extracellular DNA (e-DNA), and the control of cell death and motility are all factors regulated by c-di-GMP and are critical for three-dimensional biofilm structure development. c-di-GMP regulates bacterial transcription, enzyme activity, and even the function of bigger cellular structures after attaching to a number of cellular receptors. Bacterial biofilm generation and dissemination are tightly regulated processes that are influenced by genetic and environmental factors [19].

Finally, tiny non-coding RNA molecules, or sRNAs, have been found to play a role in post-transcriptional gene regulation in bacteria, involving metabolic activities, stress adaption, and microbial disease [20].

The aforementioned flowchart describes a few bacterial species that have the capability of causing biofilm-associated chronic infections in the body, and those are almost resistant to a bunch of antibiotics, resulting in the emergence of antimicrobial resistance and difficulties in treating the diseases [21,22,23,24,25,26] (Figure 1).

## 3. Biofilm Development

Biofilm is defined as an integrated congregation of microbes that are firmly attached to a surface and incorporated into an extracellular polymeric substances (EPS) matrix. Exopolysaccharides, nucleic acids, proteins, lipids, and other biomolecules form the EPS. The emergence of immobile biofilm communities involves multifaceted events in which EPS play critical structural and functional roles that are required for biofilms to emerge. Microbial adherence to abiotic surfaces is aided by EPS. The EPS matrix also offers mechanical stability and complex chemical microenvironments, both of which are essential for the biofilm lifestyle [27].

Additionally, EPS improve biofilm tolerance to antimicrobials and immune cells. The maturation of biofilm requires several developmental stages, which eventually result in the growth of the biofilm on the surfaces. The various stages associated with the developmental phase have their own characteristics and features, which are of high importance in the case of treating the biofilm infections. In addition to this, the strategies for each growing stage require a precise understanding for the better application of the antibiotics against the biofilm. Each stage is elaborated upon in this review from the perspective of the opportunities for therapeutic interventions during the process (Figure 2). 

This section offers an overview of the involvement of sRNAs in biofilm growth, a field of study that is still in its early stages. sRNAs play a key role in the regulatory networks that control the shift to a surface-attached lifestyle. Small non-coding RNAs appear to regulate the regulatory networks governing bacteria’s decisions to connect and form biofilms vs. continue as planktonic cells, according to new data (sRNAs) [28]. This is performed via sRNAs’ finetuning regulatory networks to allow concentration-specific responses by sequestering, antagonizing, or activating regulatory proteins in response to environmental signals, or by directly influencing protein synthesis to favour or disfavour biofilm formation. The role of sRNAs in biofilm formation control is mediated by two mechanisms: I.sRNAs functioning by base-pairing with other RNAs;II.Protein-binding sRNAs mimicking the protein binding regions found in various mRNAs to oppose and sequester their cognate regulatory proteins.

Several sRNAs have been discovered that alter the expression or activity of transcriptional regulators and the regulatory network components necessary for biofilm development and adhesion [29].

### 3.1. Initial Adhesion

The initiation of biofilm formation begins with the adhesion of the bacterial cells to any interface or medical device surfaces. Various signalling pathways help the bacteria to recognize the surface and initiate the proliferation, once the desired atmosphere is achieved. They typically produce an enzyme called adhesins that help the microorganisms to adhere themselves to the host. This stage clearly targets the attachment process between the organism and surfaces. Therefore, in this stage, therapeutic intervention can be performed by completely disrupting the adherence mechanism of the microbes with the surface by targeting the cell surface-associated enzymes called adhesins. An interruption between the microbes and surfaces can be achieved through this method and the initial formation of biofilm can be ceased [22].

Adhesion is the foremost parameter required for biofilm formation. The initiation and dispersal of the biofilm begins with the adhesion capacity of a particular species in relation to the surface. The use of surface adhesion along with the development of biofilm formation is the survival strategy of the bacteria that usually anchor themselves in a specific way so as to acquire nutrition from the environment. Two distinct phases of adhesion are important before the beginning of biofilm formation (Figure 3). They are primary or reversible adhesion and secondary or irreversible adhesion. These phases are completely controlled by the different expressions of the genes. The prime rationale behind this feature of surface adherence in bacteria is because the concentration of the nutrients is higher near the solid surfaces. This feature of bacteria plays out in two ways, and either can be beneficial or disastrous. Practically, any sort of surfaces is meant to be the surface for showing the adherence activity of bacteria. The surfaces can be abiotic as well as biotic in nature. The adhesion processes to take place and the number of elements that are taken into consideration include bacterial species, environmental status, the composition of the surface and the gene products. The primary adhesion with the abiotic surfaces occurs with hydrophobic interactions, whereas when the adhesion happens with biotic surfaces, it is mediated by particular molecular interactions. Below, Figure 3 describes the distinct phases of adhesion [25,26]. As discussed above, the non-specific interactions are found between bacteria and abiotic elements, but the molecular interactions are more prone in adhesions, as these adhesions happen to occur between bacteria and living elements. The different phases of adhesions seen in the case of molecular interactions are described below broadly. 

### 3.2. Early Biofilm Formation

Early biofilm formation occurs when bacteria begin to divide and create extracellular polymeric substances (EPS), which improves adherence, while also producing the matrix that encases the cell. This characteristic of biofilm can be used as part of a therapeutic intervention by targeting the production of extracellular polymeric substances along with the cellular division of the organism.

### 3.3. Biofilm Maturation

Biofilm maturation, during which 3D structures form wherein the EPS matrix serves as a multifunctional and protective substratum, allows for the formation of diverse chemical and physical microhabitats in which microorganisms prevail within polymicrobial and social interactions. Physical removal, the destruction of the EPS matrix, targeting the creation of pathogenic microenvironments (low pH or hypoxia) and social interactions (in polymicrobial biofilms), as well as the elimination of dormant cells, might all be used to disrupt the established biofilms that manifest, giving numerous opportunities for therapeutic intervention [26].

### 3.4. Dispersal

To convert the sessile biofilm into a motile form, microbial cells begin to disperse in a natural way during this phase. Bacteria that do not generate extracellular polysaccharides, on the other hand, scatter directly into the environment when mechanical force is applied. Microbial communities create several saccharolytic enzymes during dispersion, which release surface microorganisms into a new location for colonization. N-acetyl-heparosan lyase is produced by *E. coli*, hyaluronidase is produced by *Streptococcus equisimilis*, and alginate lyase is produced by *Pseudomonas aeruginosa* and *Pseudomonas fluorescens.* Flagella development occurs when microbial cells upregulate protein expression, and the process allows bacteria to travel to a new location, assisting in the spread of diseases. EPS matrix remodelling or the activation of dispersal pathways might cause biofilm dispersion, which can help in controlling the biofilm growth [25].

## 4. Molecular Interactions

### 4.1. Primary Bacterial Adhesion: Docking

Initially, the organism must be driven near to the surface, either randomly (for instance, by a flow of fluid streaming over a surface) or in a directed manner via chemotaxis and motility. The total sum of attracting or repulsive forces exerted between the two surfaces determines adhesion after the organism achieves critical closeness to a surface. Electrostatic and hydrophobic interactions, steric hindrance, van der Waals forces, temperature, and hydrodynamic forces, to name a few, are all examples of these forces. Because most germs and inert surfaces are negatively charged, electrostatic interactions tend to favour repulsion [30]. It is the characteristics of a niche, such as pH, ionic strength, and temperature, that determine whether bacteria will be attracted or repelled toward a contact surface. Similarly, the properties of the medium, and the composition of bacterial cell surfaces, affect how fast the bacteria will approach or move away from the contact surface. Adhesion is mediated by extracellular adhering appendages and adhesins are released once they encounter the surface.

However, initial attachment is dynamic and reversible, and bacteria can disengage and re-join the planktonic population if agitated by hydrodynamic pressures, repulsive forces, or nutrition availability [11,31].

### 4.2. Surface Conditioning

However, it is vital to recall that primary contact usually occurs between an organism and a conditioned surface, and the latter’s hydrophobicity varies substantially depending on the molecules in the conditioning film [32].

### 4.3. Secondary Bacterial Adhesion: Locking

The anchoring or locking phase of adhesion is characterized by molecularly mediated interactions between specific adhesins and the surface. Exopolysaccharides that combine with surface materials and/or receptor-specific ligands present on pili, fimbriae, and fibrillae, or both, help loosely attached organisms consolidate the adhesion process [33]. *P. aeruginosa*, a pathogen with a proclivity for forming biofilms, uses various attachment organelles to adhere to a surface in an irreversible manner. To wade through the liquid interface and contact the surface, retain adherence, and move across the attachment plane, *P. aeruginosa* adopts type IV pili-mediated twitching motility [34].

## 5. Physio-Chemical Surface Properties Controlling Biofilm Formation

Biofilm formation is a strenuous process, which includes specific physiochemical aspects for better formation. A number of parameters are considered for biofilm formation. The figure below describes the parameters briefly (Figure 4). Any and all physiochemical surface properties are considered to be an important element for studying the various possibilities of administering the antimicrobial agents on to the biofilm. A few of the properties have been elaborated upon below, describing the role of each one in biofilm formation.

### 5.1. Surface Charges

Surface charge is known to affect biofilm development and plays a crucial role in defining the attractive and repulsive forces between bacteria and the surface. Because most bacterial cells are negatively charged, a positively charged surface is more susceptible to bacterial adhesion, whereas a negatively charged surface is more resistant. *Pseudomonas aeruginosa* biofilms are generated on negatively charged poly(3-sulphopropylmethacrylate) surfaces and contain more cyclic diguanylate monophosphate than biofilms developed on positively charged poly(2-(methacryloyloxy)-ethyl trimethyl ammonium chloride) surfaces (Figure 5) [35]. 

### 5.2. Hydrophobicity and Hydrophilicity

Bacterial adhesion can be either increased or prevented by adjusting the hydrophobicity of a surface. Biofilm forms less frequently on hydrophobic surfaces than on hydrophilic surfaces. The hydrophobicity of the cell surface is important for adhesion to and dissociation from surfaces. In medicine, bioremediation, and the fermentation industry, cell surface hydrophobicity has both negative and beneficial effects on microbe adhesion to biotic and abiotic surfaces. Surface damage is caused by hydrophobic microbes forming biofilms; on the other hand, they can readily gather on organic contaminants and breakdown them. Microorganisms that are hydrophilic also play a role because of their great resistance to hydrophobic substances, hydrophilic microbes aid in the removal of organic waste from the environment [36].

### 5.3. Roughness and Topography

Because of the increased contact area between the material surface and the bacterial cells and the protection from shear pressures, an increase in surface roughness increases bacterial adhesion. However, the precise effects of surface roughness on bacterial adhesion and biofilm formation vary depending on bacterial cell size and shape, as well as other environmental conditions. With increasing surface roughness and better cell adherence to rougher surfaces, adhesion forces rise. *E. coli* and *Proteus mirabilis*, for example, elongate into filaments, increasing flagella surface density and increasing their flexibility and adhesion potential on rough surfaces [36].

### 5.4. Stiffness

It is becoming obvious that material stiffness has an impact on bacterial adherence and associated cell function. It is still unclear how surface stiffness and topography (apart from roughness) affect bacterial adherence and biofilm growth (Figure 6) [37,38].

## 6. Environmental Conditions

Biofilm formation is largely influenced by the environment, and the processes by which individual cell gene expression influences biofilm development have piqued researchers’ curiosity. Although the mechanisms involved in the process of biofilm formation differ depending on the features of the bacterial strain and environmental conditions, it is now well established that bacteria may cling to practically any surface and create biofilms (Figure 7).

### 6.1. pH

Bacterial colonization is thought to play a role in the shift to an alkaline pH. Pathogenic bacteria have shown that they prefer a more alkaline environment to develop in, and that a higher pH encourages bacterial colonization and proliferation. As a result, when the tissue is exposed, it gives a chance for the local skin flora to colonize the wound. As the wound progresses into a chronic wound, the pH shifts, making the wound environment alkaline. Finding a means to restore the skin’s acidic environment would efficiently reduce the microbial burden on the skin’s surface and lower the risk of bacterial colonization in chronic wounds [37].

### 6.2. Temperature and Moisture Content

The rate of microbial activity, as well as the proliferation of biofilms and the settling of organisms in aquatic systems, is known to be influenced by temperature. Biofilm development was induced and inhibited at different temperatures and pH levels, as well as with varying quantities of glucose [38].

### 6.3. Nutrient Availability

One of the most important factors controlling microorganism growth and activity is nutrient availability. Biofilm bacteria obtain nutrients by collecting trace organics on surfaces via extracellular polymers, utilizing waste products from neighbours and secondary colonizers, and breaking down their food supply with various enzymes. Bacterial cells within the biofilm have a lot of food in comparison to the rest of the environment. Carbon supply, the amount of nitrate, phosphate, calcium, and magnesium, as well as the impacts of osmolarity and pH are all elements to consider [37].

### 6.4. Microbial Products

The communication between nearby bacterial cells via signal molecules is aided by quorum sensing. This is a social behaviour that allows mono- and mixed bacterial colonies to interact. Quorum sensing is based on the release and production of signal molecules known as autoinducers, and it increases as cell density increases, though physiological conditions may also play a role. In terms of induction vs. repression of biofilm formation, quorum sensing plays a significant role in the development of biofilms, though it varies depending on the bacterial species and environmental conditions. Peptides are frequently used as autoinducers in Gram-positive bacteria. The aforementioned quorum-sensing molecules, antimicrobial peptides, and exopolysaccharides are the microbial products that adversely affect biofilm formation [37].

## 7. Emergence of Mixed Biofilm

In nature, most biofilms are generated by numerous microbial species, and so forth mixed-species biofilms mimic microorganisms’ true lives, including bacteria, fungus, viruses (phages), and protozoa. The supply of water, the food service industry, and human and animal health are all threatened by mixed-species biofilm development and environmental resilience. Mixed-species biofilms are thought to be responsible for 60–80% of microbial illnesses. The physiological events that occur during mixed-species community cell proliferation and biofilm development are complicated. Exterior treatments from hostile environments cause high levels of resistance and physiological changes in microorganisms, which are prominent difficulties in biofilm control. Biofilms contain microbial cells incorporated in extracellular polymeric substances (EPS), and they account for roughly 90% of the biofilm volume [39]. The “one organism—one disease” approach is used in traditional diagnoses and treatment regimens for infections, including chronic wounds. Chronic infections, on the other hand, frequently contain a large number of species in biofilms, but interactions across individual organisms are poorly known. Although the impact of bacterial population and interspecific interactions on the severity and treatment of chronic infections is unknown, recent experimental investigations show that the presence of two or more species increases resistance and virulence when compared to single-species infections. In mixed-species biofilms, cooperative microbial relationships may be based on improving the adherence of the secreted matrix produced by partners or metabolic cross-feeding with products that support the growth of other members [40]. Microbes in mixed-species biofilms may potentially help to boost resistance. For the physiological contribution of the entire consortium, the arrangement of microorganisms inside mixed-species biofilms is delicately controlled. Mixed-species biofilms can create an intermixing structure, whereas single-species biofilms lack commensal interactions between species. Due to the shift in the makeup of the biofilm matrix and the increased microbial interactions in the consortia, the major mechanisms of higher resistance among mixed-species biofilms are not totally known. Several explanations have been presented to describe the increased resistance in mixed-species biofilms. For instance, certain species may shield others by aggregating with other strains within a three-dimensional structure physically constructed by other species. The thickness of the matrix is thought to have a role in mixed-species resistance in biofilm by preventing risk factors from reaching the active bacteria’s lower levels. Interspecies interactions, such as antimicrobial resistance genomic swap, antimicrobial enzymatic transition, quorum-sensing signal-induced gene expression changes, and metabolite-mediated electron transport chain inhibition, can alter the physiology of neighbouring species in a mixed-species biofilm. These pathways could lead to mixed-species biofilms being more resistant to antibiotics than planktonic cells, as well as the therapeutic challenge of chronic biofilm-associated infections in the host. Various interactions, such as bacteria–bacteria, bacteria–fungus, and bacteria–virus, form the mixed biofilms that are responsible for the chronic infections, or any other biofilm-associated infections, in the hosts [41] (Figure 8).

### 7.1. Bacteria–Bacteria Mixed Biofilms

Biofilm production may result from the bacterial colonization of chronic wounds, which causes local and systemic inflammation and slows the healing process. Multiple bacterial species commonly inhabit chronic wounds, with *S. aureus* and *P. aeruginosa* being the most prevalent. Lung infections, particularly in cystic fibrosis, are frequently polymicrobial. Periodontal disease is another remarkable example of mixed-biofilm-associated bacterial infection. This term refers to a variety of clinical symptoms ranging from moderate, reversible gingivitis to severe, chronic periodontitis, which can result in the progressive degradation of bone and connective tissue in the periodontal area, as well as tooth loss [42].

### 7.2. Bacteria–Fungi Mixed Biofilms

Bacteria and fungi frequently coexist in same ecological and bodily habitats, generating polymicrobial biofilms and developing interkingdom interactions that are progressively being recognized as important in the aetiology of many illnesses. Bacteria–fungi mixed infections can arise anywhere around the body, including the skin, the mouth, the lungs, the gastrointestinal system, the female reproductive tract, and the bloodstream, as a result of the mixed colonization of intravascular devices. When *S. aureus* binds in huge numbers to *C. albicans* hyphae, the bacteria become more resistant to vancomycin [43].

### 7.3. Bacteria–Virus Mixed Biofilms

Emerging research reveals that bacteria and viruses may develop a complex interplay that influences the outcome of mixed infections and the response to antimicrobial/antiviral therapy. Although bacteria do not allow eukaryotic viruses to infect them, they can aid viral fitness by improving virion stability, encouraging eukaryotic cell infection, or raising coinfection rates. Viruses that bind to bacteria, on the other hand, may enhance bacterial adhesion to eukaryotic cells. For example, in cystic fibrosis patients, *P. aeruginosa* and *respiratory syncytial virus* (RSV) are the two most common infections [44]. 

## 8. Biofilm-Associated Human Diseases

Biofilms are becoming more widely acknowledged as a factor in human disease. According to studies, 24% of adults have lost at least 4 mm of periodontal attachment, and 40–50% of adults, 60% of 15-year-olds have some sort of gingival (biofilm) infection. Biofilms are also key colonizers of medical equipment, such as catheters, shunts, and venous and arterial shunts. Thus, 15–20 percent of 4000 newborns with cerebrospinal fluid shunts were found to be infected by a biofilm in a study. Respirators, sigmoidoscopes, contact lenses, and artificial implants (such as heart valves, pacemakers, ventricular assist devices, synthetic vascular grafts, and stents) are also included. Biofilms have been found on urinary prosthesis as well as orthopaedic prostheses (such as artificial joints). The majority of untreated chronic diseases in humans are assumed to be of biofilm origin by the biofilm research community. Human biofilm infections are caused by two different properties shared by all biofilms. To begin with, biofilms are extremely resistant to immunological destruction and clearing, as well as antimicrobial treatment. Second, biofilms that are well-protected may be capable of shedding individual bacteria and sloughing biofilm fragments into neighbouring tissues and the circulatory system. These shed cells could be the source of an acute sickness that recurs after antibiotic treatment [36].

Importantly, biofilm formation as a defensive measure could have far-reaching consequences for the host, as bacteria thriving in these matrix-enclosed aggregates are more resistant to drugs and host defences. Microbial cells developing in biofilms are said to be clustered, according to the biofilm model. This casts doubt on the idea that infectious pathogens are evenly dispersed and, thus, equally susceptible to the host immune system or antibiotic therapy. It could also explain a number of difficult clinical issues, including symptomatic but unculturable inflammation, antibiotic resistance, recurrence or persistence, and metastasis or dissemination. Despite the fact that intravenous catheters, prosthetic heart valves, joint prostheses, peritoneal dialysis catheters, cardiac pacemakers, cerebrospinal fluid shunts, and endotracheal tubes save millions of lives, they all carry the risk of surface-associated infections. The *Staphylococci* (especially *S. epidermidis* and *S. aureus*) are the bacteria most frequently associated with medical devices, followed by *P. aeruginosa* and a slew of other environmental bacteria that unscrupulously infect a host who is compromised by invasive medical intervention, chemotherapy, or a pre-existing disease state. The development of biofilms on medical implants has even resulted in the identification of a novel infectious condition known as chronic polymer-associated infection [45,46].

## 9. Therapeutic Approaches to Combat Biofilm Development

Biofilm-related infections and the obstacles associated with their treatment have been identified as important dangers to human health in recent years, prompting the development of solutions to address the issue (Figure 9).

Planktonic cells can be destroyed with the help of antibiotics, antimicrobial peptides, and nanoparticles. Antibiotics and treatment options against biofilms are limited, prompting the development of innovative techniques to identify potential antibiofilm candidates. Because biofilms are resistant to antibiotics, a combination therapy using many drugs was considered to try to remove biofilms. Antibiotics have lost their effectiveness as a result of the growth of multidrug-resistant bacteria and persister cells. The most important research direction to counteract the development of bacterial resistance is still the hunt for natural high-efficiency antibiotics and the development of novel broad-spectrum antibiotics [47].

CD437 is a new synthetic retinoid antibiotic that kills methicillin-resistant S. aureus (MRSA) persister cells by disrupting their lipid bilayer, and it has an excellent synergistic antibacterial activity with gentamicin. Similarly, 12 amino acid sequences were inserted into tobramycin, an aminoglycoside antibiotic, to create Pentobra, a modified antibiotic. Researchers attempted to integrate several methods of antibacterial action by imitating the function of antimicrobial peptides with the connected amino acid sequences. E. coli and S. aureus were both resistant to the complex antibiotic. More molecules might enter persister cells and trigger bactericidal action because the polypeptide increased cell membrane permeability. Because there are no specific protein targets and bacteria are unlikely to acquire drug resistance, host defence peptides are good options for treating dormant persister cells. Peptides have low in vivo stability, involve a time-consuming manufacturing process, and are expensive, despite the fact that some peptides exhibit significant antibacterial activities. All of these factors limit the use of host defence peptides in therapeutic settings (they are also called antimicrobial peptides). Researchers have been focusing on producing peptide mimics, particularly polymeric mimics that can be made quickly and cheaply, in light of these factors. The mimics have good antibacterial effects because their structural qualities are similar to those of host defence peptides. Poly(2-oxazoline)s, which were recently discovered, can act as a new class of functional peptide mimics, demonstrating that poly(2-oxazoline)s can perform as powerful antibacterial effects as the synthetic mimics of host defence peptides. The discovery of peptide mimics opens up new possibilities for antibacterial drug development [48,49].

Mass sensing is a microorganism-to-microorganism communication system that regulates the expression of genes by reacting to the density of the population and the proximity of other microbes.

When small chemicals called self-inducers or quorum-sensing molecules aggregate during cell development, they cause changes in microbial gene expression. Gram-negative bacteria use N-acyl homoserine lactones (3-oxo-dodecanoyl- and butanoyl-HSL) and the quinolone signal (2-heptyl-3hydroxy-4(1H)-quinolone) as intra-species signalling molecules. In the extracellular environment, acylase can cleave the amide bond of acyl-homoserine lactones (AHLs), a quorum-sensing signal released by Gram-negative bacteria. EPS substrate proteins and adhesins can be hydrolysed by proteases. The carrier is a protease-functionalized nanogel. The ciprofloxacin encapsulation inside the nanocarrier dispersed the *S. aureus* biofilms and decreased the viability of bacterial cells within the biofilms. When used to coat medical implants, many AMPs have shown efficacy in killing cells in biofilms, interfering with EPS generation and stability, blocking QS-dependent biofilm formation, or reducing microbial adhesion. Researchers have developed physical ways to degrade EPS and increase drug diffusion in biofilms in recent years, which appears to be direct and effective. Some methods of utilising nanoparticles to produce holes or artificial channels in EPS may be more favourable, and these minor physical alterations to EPS may considerably lessen the human body’s adverse effects. Within bacterial biofilms, laser-induced vapour nanobubbles can improve medication transport and efficiency. Antibiotics’ capacity to eradicate biofilms can be improved by using magnetic nanoparticles to build artificial channels in biofilms. Magnetic iron oxide nanoparticles (MIONPs) formed artificial channels in biofilms under the influence of a magnetic field, allowing for increased antibiotic penetration and bacterial death [50,51,52].

Natural compounds have been demonstrated to have antibacterial and anti-biofilm activity against *P. aeruginosa*, including a metabolite generated by the endophytic bacterium *Streptomyces ansochromogenes*. Maipomycin A (MaiA) is a new antibiofilm chemical isolated using a bio-guided assay from *Kibdelosporangium phytohabitans* XY-R10. MaiA is a promising option in the fight against Gram-negative biofilms. Water-borne illnesses are caused by pathogens that can be found in planktonic form or as biofilms in water-carrying pipelines. Marine mussel protein-based adhesives, which include the catecholic amino acid 3,4-dihydroxyphenylalanine (DOPA), may stick to nearly any substrate. Polydopamine (PDA), a new catechol derivative and dopamine-based coating material, has also been developed [53].

## 10. Prevalence of Antibiotic Resistance in Biofilm

Studies have proved that when bacteria are dispersed from a biofilm, antibiotic susceptibility is usually soon restored. The quick reversal of resistance after a biofilm is dispersed suggests an adaptive resistance mechanism rather than a genetic change. Therefore, a few hypotheses have been made to study the resistance in biofilm mechanisms against antibiotics.

I.The penetration of the antibiotic is either slow or incomplete.II.Concentration gradients of a metabolic source or product results in zones of bacteria that are slow or do not develop.III.Some cells show signs of an adaptive stress response.IV.Only a small percentage of cells mature into a highly protected persisted state [54].

The antibiotic treatment of nosocomial infections and wound infections is causing the evolution of antibiotic-resistant bacteria, and worries about antibiotic overuse are mounting. *P. aeruginosa* alone is the microbe that is accountable for causing wound infections in burn patients, and a survey has shown that 95% of infections are caused by *P. aeruginosa*, which are resistant to gentamicin, carbenicillin, co-trimoxazole, ceftizoxime, and tetracycline. These findings highlight the need for other anti-infective techniques, such as novel medicines and vaccines, to combat *P. aeruginosa* and other diseases. The biofilm was affirmed to be resistant when various antibiotics were tested for their efficacy on the bacteria. Some of them are cloxacillin, erythromycin, tetracycline, oxytetracycline, ceftiofur, ampicillin, streptomycin, enrofloxacin, penicillin G, trimethoprim–sulphadoxine, gentamicin, and tilmicosin. However, all the mentioned antibiotics were found to be effective extensively against the planktonic form of the microorganisms; nevertheless, they were found to ineffective against the biofilm-associated infections caused by major microbes such as *Actinomyces pyogenes, Corynebacterium renale, C. pseudotuberculosis, Staphylococcus aureus, S. hyicus*, and *Streptococcus agalactiae*. The bacteria’s ability to proliferate in biofilms is one of the main reasons for their persistence. Complement activation was lower in biofilm-grown *P. aeruginosa* than in planktonic bacteria. Additionally, bacteria aggregation into EPS-coated biofilms may render them more resistant to phagocytosis. Certain components of the humoral immune systems have also been reported to be resistant to biofilm bacteria.

Biofilm persistence causes injury to the host because phagocytic cells unleash their oxidative burst indiscriminately, inducing collateral tissue damage [46,55].

## 11. Drug Delivery Strategies to Combat Biofilm-Associated Diseases

Nanoparticles (NPs) are nanoscale unit particles. The high surface area-to-volume ratio of the nanoparticles correlates with their antibacterial activity. This is due to the fact that tiny particles have the highest surface area, which allows them to interact with bacteria more effectively and boost their antibacterial properties. The composition and dimensional structure of NPs are used to classify them. These classifications include carbon-based NPs (carbon nanotubes, fullerene, and graphene), metallic-based NPs (silver, gold, copper, iron, arsenic, zinc, nickel, chromium, and so on), natural polymeric NPs (chitosan, hyaluronic acid, and albumin), synthetic polymeric nanoparticles (poly(glycolic acid), acrylic acid, and poly(lactic acid)), and dendrimers (metal matrix nanocomposites, polymer matrix nanocomposites, and ceramic matrix nanocomposites) [56]. NPs have gained popularity as an alternative to antibiotics in the treatment of infections caused by a variety of bacteria. Because NPs have extremely good bactericidal or microbicidal implications against microbial species, such as bacteria, by associating directly with the bacterial cell wall without perforating the cell, they are considered to be a leading promising treatment to mitigate bacteria or other microorganisms that have developed resistance to antimicrobial drugs [57].

In addition to inorganic carriers, such as metal nanoparticles, successful delivery systems have used lipidic- and polymeric-based formulations, such as liposomes and cyclodextrins, respectively. Because of their enticing strength and good biocompatibility with host cells, biodegradability, harmfulness to pathogenic microbes, quench toxicity effects, and controlled release of drugs for the treatment of several diseases, polymeric-based NPs have shown great potential for targeted drug delivery in the biomedical industry. Metal-based NPs, unlike antibiotics, directly target the cell wall, break and damage the DNA structure, impede enzyme function, and stop a bacterial cell’s biological activities. Metallic-based NPs use three basic methods to exert their antibacterial action, according to theory. Targeting the bacterial cell membrane (phospholipid bilayer), protein and DNA disruptions, ATP depletion, and ROS creation are some of the strategies used [58,59,60].

## 12. Future Perspective

While our knowledge of biofilm microenvironments continues to grow, technological advancements are opening up previously unimagined possibilities for developing multi-targeted therapeutic approaches to prevent and disrupt biofilms or improve drug efficacy. Among them, surface modifications, physical removal systems, and nanoparticles are considered to be efficient new technologies against biofilm control. Surface anchoring, or the integration of an antibiotic or biocide within a reservoir coating, has long been investigated as a way to prevent bacterial adherence and biofilm formation. Material and surface engineering advancements have resulted in the development of well-defined topographic surface patterns that can inhibit biofilm formation without the use of antimicrobial agents. Surface charge, hydrophobicity, roughness, topography, and chemistry, among other material and surface attributes of medical devices, can be adjusted to prevent bacterial attachment and, hence, mitigate or block biofilm formation. Furthermore, stimuli-triggered responsive surfaces can be built that only have an effect when they come into contact with cell wall-associated or membrane-associated adhesins or bacterial chemical signals. New biofilm removal technologies, such as mechanical, energy-based, and light-based disruption, may help to improve biofilm intervention tactics. Considering the complexity of biofilm development and microbial interactions with the physical and chemical environment, a balance of these techniques may be required to effectively address biofilm-mediated illness [61].

Progresses in nanoparticle production have led to the development of a variety of biofilm-fighting strategies. Inorganic metallic nanoparticles and organic nanoparticles are being studied more and more to increase their anti-biofilm performance and biocompatibility in order to reduce host toxicity. Nanoparticles can be utilized to create nano-coatings, which can be incorporated into materials as composites or fillers, or mixed with antimicrobials and other methods to physically disrupt or eradicate biofilms [62]. Liposomes are vesicles made up of one or more phospholipid bilayers that are physiologically acceptable, and they are one of the most extensively used organic nanoparticles for drug delivery.

They have good biofilm penetration, are biocompatible, and exhibit efficacy against biofilms of a variety of bacterial species for a variety of antibiotics. These nanocarriers can protect the therapeutic drug from harmful interactions with the matrix, as well as enzymatic degradation and destruction by other bacterial and host components at the infection site. The lipid structure can also combine with the bacterial outer membrane, allowing the medicine to be released directly into the cell, potentially increasing therapeutic effects while decreasing host cytotoxicity. Liposomes can also be functionalized by connecting biomolecules, for example, peptides and pH-responsive polymers on the nanoparticle surface, to boost targeting selectivity and trigger release. Nanoparticles activated the formation of free radicals from H_2_O_2_ in situ under acidic (pathological) conditions, causing the biofilm matrix to degrade and the attached bacteria to die quickly [62,63,64,65,66,67,68,69,70] (Figure 10).

## 13. Concluding Remarks

The surface, the microbe, and the EPS all interact in a complicated and dynamic way to start a biofilm. After a biofilm has formed, its adhesive strength and viscoelastic qualities make it difficult to remove it from surfaces, and resident microorganisms grow resistant to antimicrobials.

Despite the fact that tolerance is a prevalent feature of biofilms, the mechanisms behind tolerance as a microbial survival strategy are complex. Similarly, we believe that a reciprocating multidimensional approach to biofilm reduction is considerably more likely to yield therapeutic success than a frantic search for a miracle cure. Inducing biofilm-disruptive activity in response to pathogenic microenvironments could be a critical strategy (for example, acidic pH, hypoxia, or pathogen-derived metabolites). As a result, the biological effects can be fine-tuned to target the biofilm microenvironment, destroy the matrix, and kill resident bacteria, removing the pathogenic niche with accuracy and little damage to adjacent tissues. Nonetheless, a disconnect between new technology research and commercialization initiatives was evident. Clinical researchers will be able to examine the efficacy of these new technologies in clinical trials owing to a collaborative effort of chemists, engineers, and biomedical researchers, as well as toxicity and safety investigations. Through this article, we can conclude that to treat all these biofilm-associated infections, a highly efficient treatment will involve establishing a collaborative multidisciplinary workflow, focusing on the removal of the contaminated foreign bodies, the selection of biofilm-active, sensitive, and well-penetrating antibiotics, topical antibiotic delivery in high doses and combinations, and anti-quorum sensing or biofilm dispersion agents, which are all part of a multidisciplinary partnership. This is the ultimate approach of One Health, which has been discussed in this article as a concept for a therapeutic strategy. The treatment of biofilm infections currently necessitates interdisciplinary collaboration in clinical microbiology, surgery, internal medicine, pharmacology, and basic research. The removal of infected indwelling devices, the selection of highly penetrating and sensitive antibiotics, and the early administration of high-dosage antibiotics in combination and supplemented with anti-QS treatment and/or biofilm dispersal agents are all things we believe should be included in biofilm treatment right now [8]. Future research should concentrate on maximizing efficacy and specificity while minimizing toxicity and long-term therapeutic effects, as well as forming industrial partnerships to generate low-cost and viable commercial formulations.

## Figures and Tables

**Figure 1 antibiotics-11-00476-f001:**
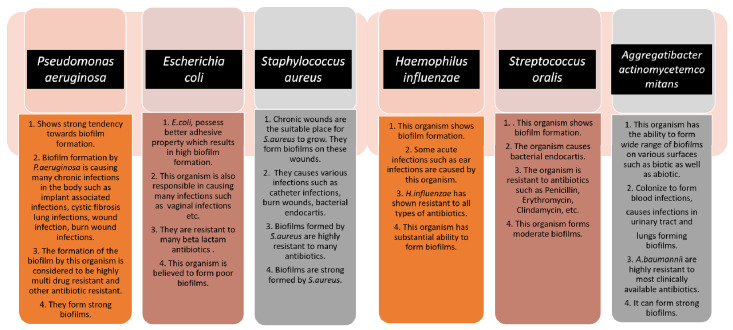
The schematic diagram highlights the characteristics of each microorganism in terms of biofilm formation and the diseases caused by them. The strength of the biofilm formation and antibiotic resistance are also mentioned.

**Figure 2 antibiotics-11-00476-f002:**
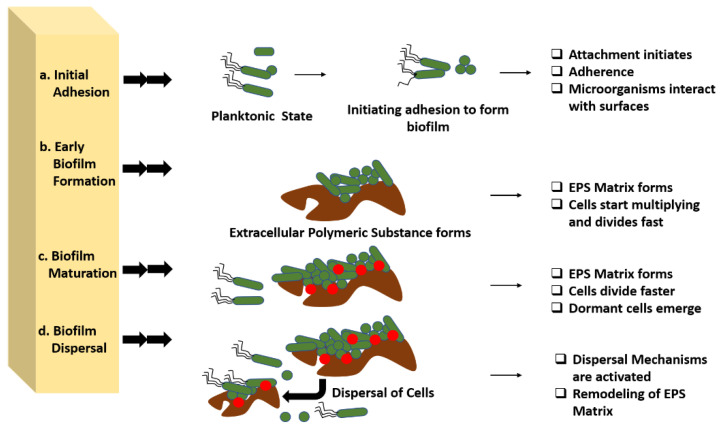
Developmental stages of biofilm and various opportunities for antibiotic interventions in these stages. Four distinct stages are present in biofilm development that can be used for various therapeutic interventions. (**a**) Initial adhesion: in this stage therapeutic intervention can be performed by completely disrupting the adherence mechanism of the microbes with the surface by targeting the cell surface-associated enzymes called adhesins. (**b**) In early biofilm formation, the biofilm can be targeted in the production of extracellular polymeric substances along with the cellular division of the organism. (**c**) Biofilm maturation, physical removal, destruction of the EPS matrix, and targeting the creation of pathogenic microenvironments can be performed for treating the biofilm maturation. (**d**) Biofilm dispersal, EPS matrix remodelling, or the activation of dispersal pathways might cause biofilm dispersion, which can help in controlling the biofilm growth.

**Figure 3 antibiotics-11-00476-f003:**
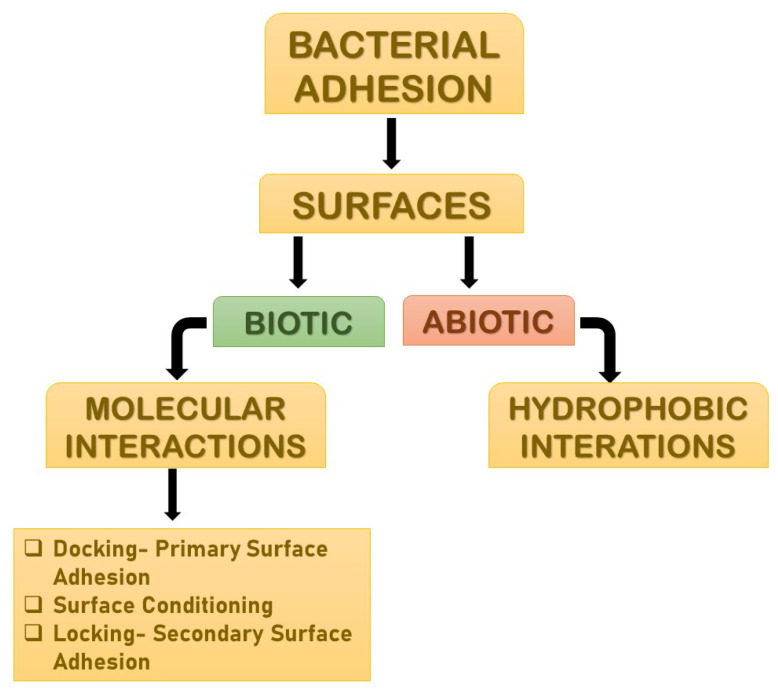
The distinct phases of adhesions. They are mainly primary or reversible adhesion and secondary or irreversible adhesion. These phases are completely controlled by the different expressions of the genes. The reason behind this feature of surface adherence in bacteria is because the concentration of the nutrients is higher near the solid surfaces. This feature of bacteria plays out in two ways, and either can be beneficial or disastrous.

**Figure 4 antibiotics-11-00476-f004:**
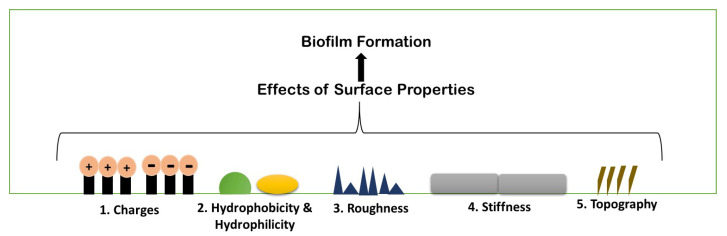
Schematic illustration of different existing properties of surface or biofilm formation.

**Figure 5 antibiotics-11-00476-f005:**
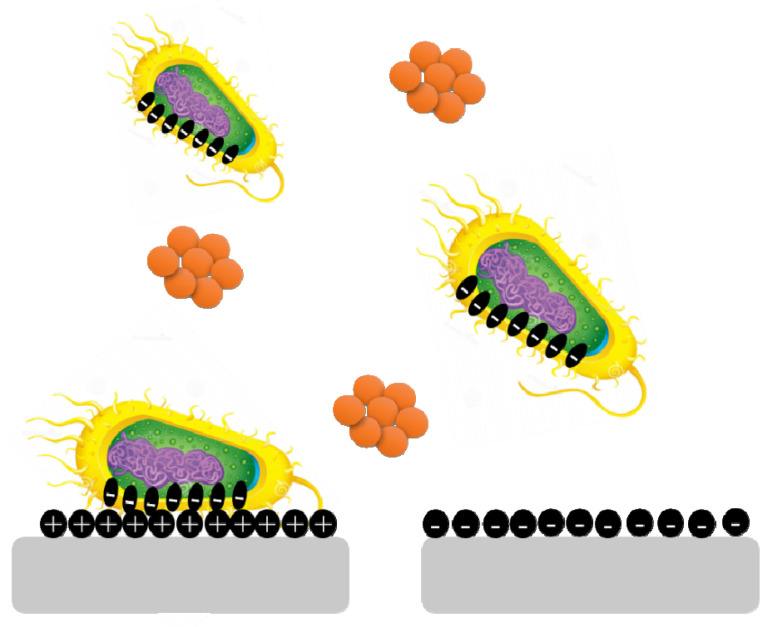
Bacterial cells are negatively charged, and a positively charged surface is more vulnerable to bacterial adherence than a negatively charged surface.

**Figure 6 antibiotics-11-00476-f006:**
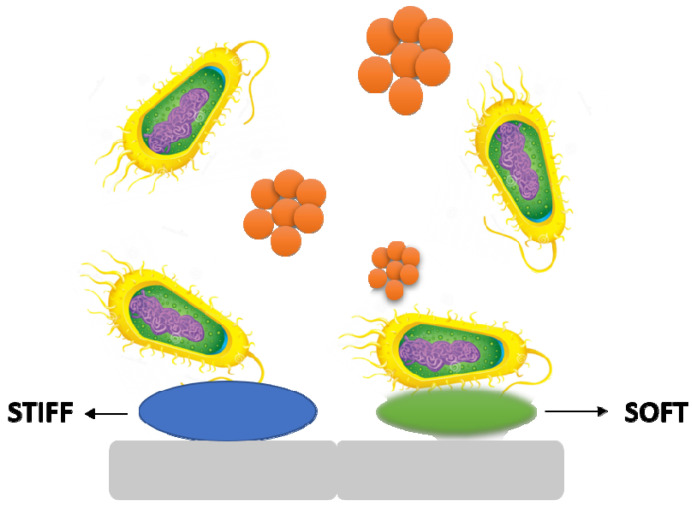
The stiffness of a material has an effect on bacterial adhesion and cell activity.

**Figure 7 antibiotics-11-00476-f007:**
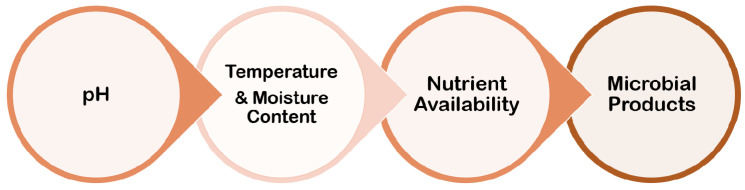
Several parameters related to environmental conditions for biofilm formation.

**Figure 8 antibiotics-11-00476-f008:**
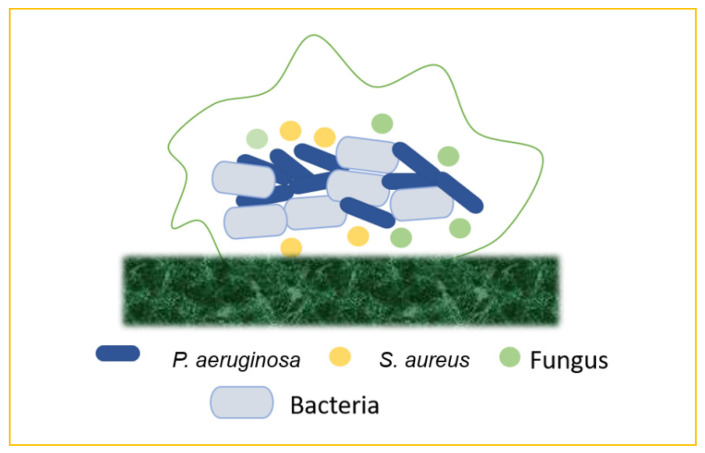
Mixed biofilms (bacteria–bacteria; bacteria–fungus; bacteria–virus).

**Figure 9 antibiotics-11-00476-f009:**
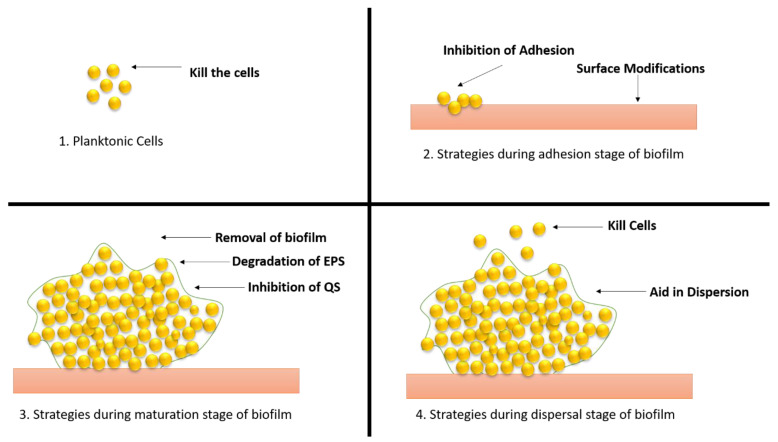
Different strategies of therapeutic approaches to combat the biofilm development. (**1**) Strategies applied to planktonic cells; (**2**) approaches applied during adhesion stage; (**3**) approaches applied during maturation stage; (**4**) approaches applied during dispersal stage.

**Figure 10 antibiotics-11-00476-f010:**
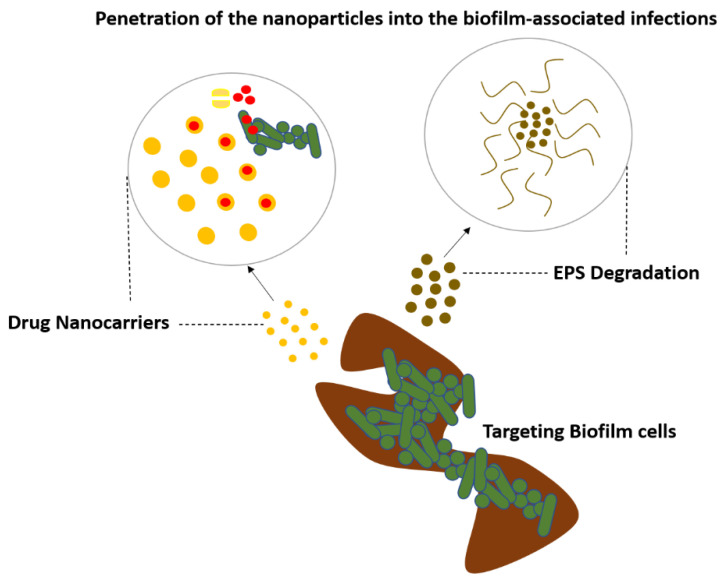
Nanoparticles targeting the biofilm to treat the biofilm-associated infections.

## Data Availability

Data available in a publicly accessible repository. The data presented in this study are openly available in the references and manuscript.

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
