# Peer review of "Emerging Concern with Imminent Therapeutic Strategies for Treating Resistance in Biofilm"

_antibiotics, 2022, doi:10.3390/antibiotics11040476_

Round 1
Reviewer 1 Report
I am delighted to review this review article on biofilm-forming microbial infections therapy and structural and functional characteristics of microbial biofilms as well as drug tolerance mechanisms. The manuscript follows the scope of the journal Antibiotics. A detailed literature survey has been done with an excellent presentation, and with a proper diagram. A flow chart of the topic going to be discussed in this review at the beginning of the discussion will have a good overview for the readers.
I would recommend the article could be published in Antibiotics with minor corrections. And the authors need to address the below-mentioned queries.
(a) The author should include the introduction.
(b) Correct the font size from lines 69-72.
(c) Table 1: Either follow alphabetical order or first show the biofilm-associated diseases caused by Gram-positive followed by Gram-Negative organisms.
(d) Line 194: “Below, a figure”: Write the figure number
(e) The author could elaborate on the primary or reversible adhesion and secondary or irreversible adhesion.
(f) Show the diagram for the effect of “Surface Charge” and “Stiffness “on biofilm formation.
(g) The author could make a table to show antibiotic resistance marketed drugs associated with the biofilm-associated disease and treatment options.
(h) The author could have highlighted a bit on the topic of nanoparticles for drug delivery from recent literature.
(i) The author could include the following references.
(i) Srinivasan R, Santhakumari S, Poonguzhali P, Geetha M, Dyavaiah M, Xiangmin L. Bacterial Biofilm Inhibition: A Focused Review on Recent Therapeutic Strategies for Combating the Biofilm Mediated Infections. Front Microbiol. 2021 May 12;12:676458. doi: 10.3389/fmicb.2021.676458. PMID: 34054785; PMCID: PMC8149761.
(ii) Machado D, Castro J, Palmeira-de-Oliveira A, Martinez-de-Oliveira J, Cerca N. Bacterial Vaginosis Biofilms: Challenges to Current Therapies and Emerging Solutions. Front Microbiol. 2016 Jan 20;6:1528. doi: 10.3389/fmicb.2015.01528. PMID: 26834706; PMCID: PMC4718981.
(iii) Kostakioti M, Hadjifrangiskou M, Hultgren SJ. Bacterial biofilms: development, dispersal, and therapeutic strategies in the dawn of the postantibiotic era. Cold Spring Harb Perspect Med. 2013 Apr 1;3(4):a010306. doi: 10.1101/cshperspect.a010306. PMID: 23545571; PMCID: PMC3683961.
(iv) Wasfi R, Hamed SM, Amer MA and Fahmy LI (2020) Proteus mirabilis Biofilm: Development and Therapeutic Strategies. Front. Cell. Infect. Microbiol. 10:414. doi: 10.3389/fcimb.2020.00414
(v) Thambirajoo M, Maarof M, Lokanathan Y, Katas H, Ghazalli NF, Tabata Y, Fauzi MB. Potential of Nanoparticles Integrated with Antibacterial Properties in Preventing Biofilm and Antibiotic Resistance. Antibiotics (Basel). 2021 Nov 2;10(11):1338. doi: 10.3390/antibiotics10111338. PMID: 34827276; PMCID: PMC8615099.
(vi) Martin C, Low WL, Gupta A, Amin MC, Radecka I, Britland ST, Raj P, Kenward KM. Strategies for antimicrobial drug delivery to biofilm. Curr Pharm Des. 2015;21(1):43-66. doi: 10.2174/1381612820666140905123529. PMID: 25189862.
(vii) Mitchell, M.J., Billingsley, M.M., Haley, R.M. et al. Engineering precision nanoparticles for drug delivery. Nat Rev Drug Discov 20, 101–124 (2021). https://doi.org/10.1038/s41573-020-0090-8
(j) Correct line spacing of references 53-55.
Author Response
The reviews suggested by the Reviewer-01 have been incorporated in the manuscript. Following are the corrections made.
(a) The author should include the introduction.
- Correction incorporated.
(b) Correct the font size from lines 69-72.
- Correction incorporated.
(c) Table 1: Either follow alphabetical order or first show the biofilm-associated diseases caused by Gram-positive followed by Gram-Negative organisms.
- Correction incorporated.
(d) Line 194: “Below, a figure”: Write the figure number
- Correction incorporated.
(e) The author could elaborate on the primary or reversible adhesion and secondary or irreversible adhesion.
- Correction incorporated.
(f) Show the diagram for the effect of “Surface Charge” and “Stiffness “on biofilm formation.
- Correction incorporated.
(g) The author could make a table to show antibiotic resistance marketed drugs associated with the biofilm-associated disease and treatment options.
- various chemicals that act as antibiofilm resistance has been mentioned n the manuscript that is used for treating the diseases.
(h) The author could have highlighted a bit on the topic of nanoparticles for drug delivery from recent literature.
- Correction incorporated.
(i) The author could include the following references.
- Correction incorporated.
(j) Correct line spacing of references 53-55.
- Correction incorporated.

Reviewer 2 Report
The review of Pandei et al., titled "Biofilm resistance: Emerging concern with imminent therapeutic strategies" focuses on the role of biofilm in the pathogenesis of bacterial infections and in the consequent resistance to antibiotics that this particular architecture is able to give to bacteria organize themselves in communities.
The topic treated in the manuscript is of topicality and relevance for both the medical and pharmaceutical world, as AMR risks becoming the main cause of death starting from 2050 and for this reason the fight against AMR is in the top-four of the actions promoted by WHO. However, before considering for publication the manuscript should be extensively revised. -The first major problem encountered concerns the bibliography. Apart from the fact that the position of some references should be checked as it does not follow the ascending order and the style should be standardized according to the dictates of the journal, the main problem is that for a review on such a topical and scientifically relevant topic, they have been only 55 papers cited in total. There are entire paragraphs and sub-chapters that refer to a single cited reference, which moreover is very often another review. Not to mention that there are phrases and concepts without references. Ideally, in a review, each sentence and each statement reported should have its own reference.
Therefore, please review the bibliography extensively. -The first chapter should be revised and made more organic in the subject matter, as some concepts are only hinted at, or superfluous in the context of the paragraph, or taken up in tranches during the manuscript.
For example, introducing the reader to what biofilm is and the problem of resistance connected to it, we speak of the "vascularization of colonies". as reported, the sentence is superfluous and decontextualized. It could be moved later when talking about the organization of bacteria in biofilms and in the development stages of the biofilm itself.
-Furthermore, again in the first chapter, as we want to focus attention on the importance of biofilms in AMR, in infections for humans, and on the impact they have on a global level for which a One Health intervention is required.
For this reason this chapter should be developed better; for example paragraph 2.1 and 2.2 as well as chapter 12 could be integrated here.
-Figure 1 and Table 1 can be implemented in a single board which shows the different bacterial species capable of forming biofilms, the characteristics of the biofilms that
they are able to train and the associated pathologies, with the relative references
-Paragraph 2.3 which deals with the role of QS in biofilm formation is a bit superficial, considering the amount of scientific evidence that correlates these two phenomena. This paragraph also suffers from the complete lack of bibliographic references.
-Well the paragraphs that deal with the processes and factors that regulate membership.
-In chapter 10 which deals with the possible therapeutic approaches to combat the development of biofilms, in treating as a possible strategy that of interfering with the QS there is no reference between the various messengers to the AI-2 self-inducer (DPD) and the targets it controls as for example LsrK. Several experimental studies and reviews underline the importance of this target for the development of new anti-AMR strategies.
In general, a discussion of the possible pharmaceutical targets that are currently being studied in this area should be reported and treated for their potential
Author Response
-Furthermore, again in the first chapter, as we want to focus attention on the importance of biofilms in AMR, in infections for humans, and on the impact they have on a global level for which a One Health intervention is required.
For this reason this chapter should be developed better; for example paragraph 2.1 and 2.2 as well as chapter 12 could be integrated here.
- Correction incorporated
-Figure 1 and Table 1 can be implemented in a single board which shows the different bacterial species capable of forming biofilms, the characteristics of the biofilms that
they are able to train and the associated pathologies, with the relative references.
- The content of the table as well as in the figure describes different topics. Thus, keeping the diseases associated with the biofilm has been kept separated from the characteristics or features of the organisms forming the biofilm.
-Paragraph 2.3 which deals with the role of QS in biofilm formation is a bit superficial, considering the amount of scientific evidence that correlates these two phenomena. This paragraph also suffers from the complete lack of bibliographic references.
- Correction incorporated.
-Well the paragraphs that deal with the processes and factors that regulate membership.
- Changes done
-In chapter 10 which deals with the possible therapeutic approaches to combat the development of biofilms, in treating as a possible strategy that of interfering with the QS there is no reference between the various messengers to the AI-2 self-inducer (DPD) and the targets it controls as for example LsrK. Several experimental studies and reviews underline the importance of this target for the development of new anti-AMR strategies.
In general, a discussion of the possible pharmaceutical targets that are currently being studied in this area should be reported and treated for their potential
- Correction incorporated

Reviewer 3 Report
This review well describe the characteristics of biofilms but there is still not sufficient to explain the therapeutic strategies.
- Lines 45-46: The sentence should be revised for clarifying the statement.
- Sub-sections 2.1 and 2.2 should be combined or change the subtitle.
- Table 1 – provide more information by adding more relevant pathogens.
- Table 1 should be included in Figure 1. That should be better pictorial description.
- Figure 1 should also be classified based on the ability of pathogens to form biofilms such as strong, medium, and weak biofilm former and also the antibiotic (or other stresses) resistance profiles.
- Among the biofilm formation regulatory systems, sRNA is to be more described in detail.
- The subsection 3.1 and 4 are redundant. Adhesion should be stated in the section 3.1 or rearrange it.
- Lines 237-238: The terms attractive and repulsive should be better than the terms susceptibility and resistant.
- In Section 8, some statements really need cited references.
- Bacteria-to-bacteria mixed biofilms should be more specified in terms of interspecies and intraspecies.
- All names of bacteria need to be italicized throughout the manuscript.
- It is difficult to see the connection biofilm formation and One-Health approach.
- Current title is not suitable to cover this review manuscript because mostly biofilm formation has been described and therapeutic methods have been just stated at the end of this review manuscript, Future perspective. Then, the title should be revised or this review should be re-structured mainly based on the therapeutic approaches to control biofilms.
Author Response
- Lines 45-46: The sentence should be revised for clarifying the statement.
- Correction incorporated.
- Sub-sections 2.1 and 2.2 should be combined or change the subtitle.
- Correction incorporated.
- Table 1 – provide more information by adding more relevant pathogens.
- Table 1 should be included in Figure 1. That should be better pictorial description.
- Correction incorporated.
- Figure 1 should also be classified based on the ability of pathogens to form biofilms such as strong, medium, and weak biofilm former and also the antibiotic (or other stresses) resistance profiles.
- Correction incorporated.
- Among the biofilm formation regulatory systems, sRNA is to be more described in detail.
- Correction incorporated.
- The subsection 3.1 and 4 are redundant. Adhesion should be stated in the section 3.1 or rearrange it.
- Correction incorporated.
- Lines 237-238: The terms attractive and repulsive should be better than the terms susceptibility and resistant.
- Correction incorporated.
- In Section 8, some statements really need cited references.
- Correction incorporated.
- Bacteria-to-bacteria mixed biofilms should be more specified in terms of interspecies and intraspecies.
- Correction incorporated.
- All names of bacteria need to be italicized throughout the manuscript.
- Correction incorporated.
- It is difficult to see the connection biofilm formation and One-Health approach.
-Eventually, biofilm formation leads to antimicrobial resistance or multi drug resistance and this issue requires interdisciplinary approach which means One-Health approach.
- Current title is not suitable to cover this review manuscript because mostly biofilm formation has been described and therapeutic methods have been just stated at the end of this review manuscript, Future perspective. Then, the title should be revised or this review should be re-structured mainly based on the therapeutic approaches to control biofilms.
We have revised the title in the manuscript.

Round 2
Reviewer 2 Report
I thank the authors for having considered the suggestions of my collegues and mine. The review is suitable to be published in the present form
Author Response
Thank you so much for your appreciation.
With regards
Ramendra
Reviewer 3 Report
Additional comments are as follows;
- The term, biofilm resistance, is obscure. It should be the “resistance in biofilm” or “biofilm resistance to WHAT”.
- The term “one health” is still not suitable for this review. It would be better to be deleted throughout the manuscript.
Author Response
Dear Editor,
Thank you very much for your thoughtful feedback. The term "resistance in biofilm" has been incorporated throughout the manuscript. In addition, the one health was removed from the manuscript's suggested topic and abstract.
With Regards
Ramendra